

# Agrichemicals and antibiotics in combination increase antibiotic resistance evolution

Brigitta Kurenbach[1,*], Amy M. Hill[1,*], William Godsoe[2], Sophie van Hamelsveld[1] and Jack A. Heinemann[1]

[1] School of Biological Sciences and Centre for Integrated Research in Biosafety and Centre for Integrative Ecology, University of Canterbury, Christchurch, New Zealand
[2] Bio-Protection Centre, Lincoln University, Lincoln, New Zealand
[*] These authors contributed equally to this work.

Corresponding author
Jack A. Heinemann,
jack.heinemann@canterbury.ac.nz

## ABSTRACT

Antibiotic resistance in our pathogens is medicine's climate change: caused by human activity, and resulting in more extreme outcomes. Resistance emerges in microbial populations when antibiotics act on phenotypic variance within the population. This can arise from either genotypic diversity (resulting from a mutation or horizontal gene transfer), or from differences in gene expression due to environmental variation, referred to as adaptive resistance. Adaptive changes can increase fitness allowing bacteria to survive at higher concentrations of antibiotics. They can also decrease fitness, potentially leading to selection for antibiotic resistance at lower concentrations. There are opportunities for other environmental stressors to promote antibiotic resistance in ways that are hard to predict using conventional assays. Exploiting our previous observation that commonly used herbicides can increase or decrease the minimum inhibitory concentration (MIC) of different antibiotics, we provide the first comprehensive test of the hypothesis that the rate of antibiotic resistance evolution under specified conditions can increase, regardless of whether a herbicide increases or decreases the antibiotic MIC. Short term evolution experiments were used for various herbicide and antibiotic combinations. We found conditions where acquired resistance arises more frequently regardless of whether the exogenous non-antibiotic agent increased or decreased antibiotic effectiveness. This is attributed to the effect of the herbicide on either MIC or the minimum selective concentration (MSC) of a paired antibiotic. The MSC is the lowest concentration of antibiotic at which the fitness of individuals varies because of the antibiotic, and is lower than MIC. Our results suggest that additional environmental factors influencing competition between bacteria could enhance the ability of antibiotics to select antibiotic resistance. Our work demonstrates that bacteria may acquire antibiotic resistance in the environment at rates substantially faster than predicted from laboratory conditions.

## SIGNIFICANCE

Neither reducing the use of antibiotics nor discovery of new ones may be sufficient strategies to avoid the post-antibiotic era. This is because bacteria may be exposed to other non-antibiotic chemicals that predispose them to evolve resistance to antibiotics more quickly. Herbicides are examples of some of the most common non-antibiotic chemicals in frequent global use. We previously showed that in some combinations the herbicides we tested made bacteria phenotypically resistant to higher concentrations of antibiotics, while in other combinations bacteria became susceptible at lower antibiotic concentrations. Here we demonstrate that in both cases the herbicides worked with antibiotics to accelerate genotypic resistance evolution. Unfortunately, antibiotic resistance may increase even if total antibiotic use is reduced, and new ones are invented, unless other environmental exposures are also controlled.

## INTRODUCTION

As fundamental tools for infection control, antibiotics underpin diverse human systems ranging from hospital care to concentrated animal feeding operations through to crop and pollinator disease management. The loss of this tool due to antibiotic resistance will result in higher mortality and morbidity, but also deny access to many routine medical procedures for risk of subsequently untreatable infections (*Teillant et al., 2015*; *Thomas, Smith & Tilyard, 2014*). Antibiotic resistances also threaten agricultural productivity (*Stockwell & Duffy, 2013*; *Van Boeckel et al., 2015*). Despite over a half century of warning, neither science nor public and private innovation strategies have managed to avert the threat of a post-antibiotic era.

One stewardship strategy is reduction of use which might help increase longevity (*CDC, 2013*; *Collingnon et al., 2016*). If bacteria almost never encounter antibiotics at concentrations high enough to harm them, there would be little opportunity for resistant variants to emerge and establish. Based on this, it has been suggested that judicious and low use of antibiotics that keeps most antibiotic exposures to less than the minimum inhibitory concentration (MIC) should preserve antibiotic susceptibility in bacteria (*Andersson & Hughes, 2014*). In practice, the lowest concentration of antibiotic leading to the evolution of resistance in a given environment, the so called minimum selective concentration (MSC), can be much lower than the MIC (Fig. 1) (*Andersson & Hughes, 2014*; *Baquero et al., 1998b*; *Hermsen, Deris & Hwa, 2012*). Keeping the use of antibiotics to below MSC concentrations is more challenging still.

Variation in antibiotic responses can be caused by either genetic or physiological differences between individual bacteria. The toxic effect of an antibiotic may occur at different concentrations for different individuals because some have *acquired* genes or alleles through mutation or horizontal gene transfer (i.e., change in genotype). Also, organisms can have *innate* differences between them, e.g., due to differences in permeability.

Innate resistance can also be dependent upon genes expressed or repressed conditionally, resulting in increased efflux or decreased influx of antibiotics and overall lower intracellular antibiotic concentrations (*Fernandez & Hancock, 2012*). Such genes or expression induction

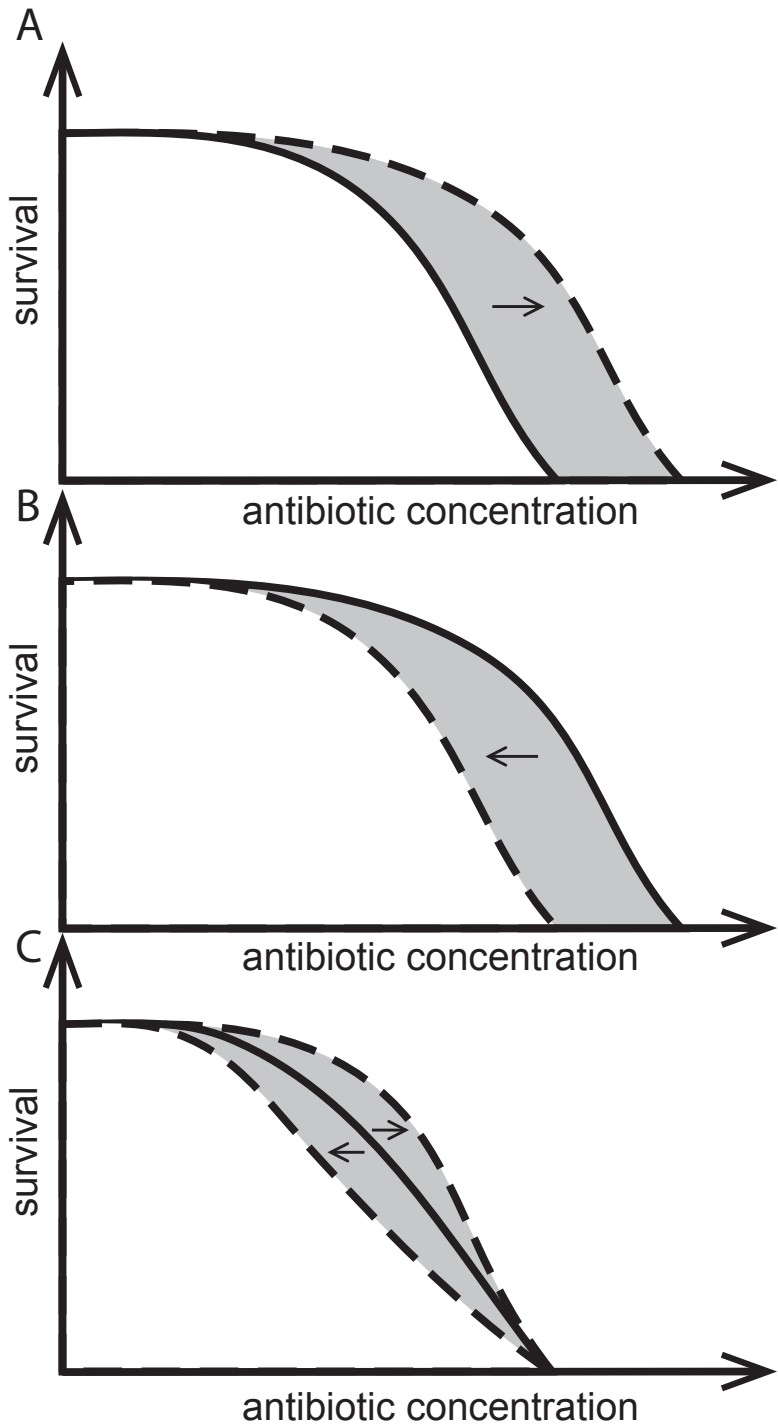

**Figure 1  Effects of herbicides on bacterial responses to antibiotics.** Herbicides can change responses to antibiotics. Solid curves indicate survival as a function of only antibiotic concentration, intersecting the *x*-axes at the MIC. Dashed curves show herbicides (A) increasing the MIC, (B) decreasing the MIC, and (C) having no effect on MIC but altering population survival at antibiotic concentrations below-MIC. The hypothetical MSC is the point of divergence of solid and dashed curves and the area in gray illustrates all antibiotic concentrations where the herbicide changes the response to antibiotics.

thresholds may differ between species, and individuals within a species may phenotypically differ depending on whether or not they expressed those genes before being inhibited by the antibiotic. This resistance through changes in gene expression is also known as an *adaptive* (change in phenotype) response (*Fernandez, Breidenstein & Hancock, 2011*) which acclimates bacteria to the environment. It can be triggered by antibiotics and other chemical toxins or environmental cues (*Blair et al., 2015*; *Palmer & Kishony, 2013*; *Sanchez-Romero & Casadesus, 2014*).

We have previously shown that concurrent exposure of *Escherichia coli* or *Salmonella enterica* sv. Typhimurium to various commercial herbicides and antibiotics from different classes can lead to adaptive increases (Fig. 1A) or decreases (Fig. 1B) in their susceptibility to the antibiotics. In some cases, herbicide exposures changed the survival at below-MIC antibiotic concentrations of antibiotic, while MIC itself was not affected (Fig. 1C). Adaptive responses were specific for the combination of bacteria, antibiotic and herbicide used. Effects were shown for commercial formulations (*Kurenbach et al., 2015*), for label listed active ingredients, and for common adjuvants (*Kurenbach et al., 2017*).

Populations of adaptively resistant bacteria can in time produce variants with acquired resistance to even higher concentrations of antibiotic (*Cohen et al., 1989*; *Gustafson et al., 1999*; *Shen, Pu & Zhang, 2011*). This raises the possibility that environmental stimuli that cause phenotypic antibiotic resistance variation between individuals could be hotspots for evolution of acquired antibiotic resistance.

Here we exploit our previous observations to test the novel hypothesis that both adaptive increases and decreases in antibiotic susceptibility caused by exposure to herbicide formulations can lead to an increase in the rate of acquired resistance evolution in populations of *E. coli* and *S. typhimurium*. For this purpose we conducted short term evolution experiments using either the natural variability within a monoculture of *E. coli* or *S. typhimurium*, or isogenic strains only differing in their level of antibiotic resistance, which represents natural variability.

## MATERIALS AND METHODS

### Media

Strains and plasmids are listed in Table 1. Bacteria were grown in standard rich growth medium, LB (Lennox) (Invitrogen) at 37 °C and supplemented with ampicillin (Amp, AppliChem), chloramphenicol (Cam, Sigma) ciprofloxacin (Cip, Pentex), streptomycin (Str, Sigma), tetracycline (Tet, Sigma), or nalidixic acid (Nal, Sigma) as appropriate. Liquid cultures were grown with aeration (180 rpm), and plates were incubated in plastic bags to avoid drying out. Commercial herbicide formulations used were Kamba[500] (Nufarm, NZ) containing 500 g/L dimethyl salt of dicamba, and Roundup Weedkiller (Monsanto, Australia) containing 360 g/L isopropylamine salt of glyphosate. Herbicide concentrations are reported in parts per million acid equivalent (ppm ae) of the active ingredient to allow for comparison with other formulations. Antibiotic and herbicide concentrations used are specified in the main text or in figure legends.
**Table 1  Bacteria and plasmids.**

| Bacteria | Genotype/comments | Relevant antibiotic resistance level | Reference/source |
|---|---|---|---|
| *E. coli* | | | |
| BW25113 | wild-type. F[−], λ[−], Δ(*araD-araB*)567, Δ*lacZ4787(::rrnB-3)*, *rph-1*, Δ(*rhaD-rhaB*)568, *hsdR514* | Cip/Cip +Kamba/Cip+Roundup[a]: Solid medium: 0.01/0.07/0.07 μg/mL Liquid medium: 0.04/0.05/0.09 μg/mL | *Baba et al. (2006)* |
| SB21 | *hsdS*, *leuB6*, *thr* | | *Heinemann & Sprague Jr (1989)* |
| AH201 (Tet[high]) | SB21 (pBR322) | Tet: 125 μg/mL | This study |
| AH214 (Tet[low]) | SB21 (pAH14) | Tet: 1.5 μg/mL | This study |
| JB436 | SB21 Nal[R] | Nal: 60 μg/mL | *Heinemann, Scott & Williams (1996)* |
| AH204 (Str[high]) | JB436 (RSF1010) | Str: 250 μg/mL | This study |
| AH211 (Str[low]) | SB21 (pAH11) | Str: 1 μg/mL | This study |
| *S. enterica* sv Typhimurium | | | |
| SL3770 | LT2, *pyr*[+], *rfa* | Cip/Cip+Kamba/Cip+Roundup[a]: Solid medium: 0.035/0.1/0.1 μg/mL Liquid medium: 0.05/0.1/0.2 μg/mL | *Roantree, Kuo & MacPhee (1977)* |
| Plasmids | | | |
| pBR322 | Amp[R], Tet[R] | Amp: 100 μg/mL, Tet: 125 μg/mL | *Sutcliffe (1979)* |
| pAH14 | pBR322 derivative, Cam[R], Amp[S], Tet[S] | Cam:20 μg/mL | This study |
| RSF1010 | Str[R] | Str: 250 μg/mL | *Scholz et al. (1989)* |
| pAH11 | RSF1010 derivative, Cam[R], Str[S] | Cam: 20 μg/mL, Str: 1 μg/mL | This study |

**Notes.**
[a]Kamba: added at 1,830 ppm ae, Roundup added at 1,240 ppm ae.

## Plasmid constructs

pBR322 (Table 1) was used as the base to create a pair of plasmids that only differed in antibiotic resistance determinants. Plasmid pAH14 was created by deleting a section of the gene for Tet[R] by removing the *Hin*dIII and *Bam*HI fragment of pBR322 and inserting *cat* from pACYC184 at the *Pst*I site within *bla*. The resulting plasmid conferred resistance to chloramphenicol, but not to ampicillin or tetracycline (Cam[R], Amp[S], Tet[S]). RSF1010 (Table 1) was the base for pAH11, which was created by insertion of *cat* from pACYC184 into the *Eco*RI and *Not*I sites of RSF1010, resulting in a plasmid conferring chloramphenicol but not streptomycin resistance (Cam[R], Str[S]).

## Culturing conditions

For the experiments described in the section "Evolution of acquired antibiotic resistance in cultures with herbicide-induced increases in MIC" *E. coli* or *S. typhimurium* were grown for 24 h in 10 mL liquid LB medium containing Cip, herbicide, both, or neither. Initial densities were ca. $10^5$ cfu/mL for the former treatment and ca. $10^6$ cfu/mL (*E. coli*) and $10^5$ cfu/mL (*S. typhimurium*) for the latter three. The experiment was abandoned at this point if cultures containing Cip but not herbicide were visibly turbid. This was because in this experiment the chosen Cip concentration was above MIC in cultures not exposed to herbicide, and growth was therefore interpreted as due to a resistant mutant in the starting culture. Herbicide and antibiotic concentrations used are detailed in the Results section. Cultures were then diluted as above in the same conditions and incubated again for 24 h. A

10-fold dilution in LB without herbicides or antibiotics followed to ensure observed effects were not due to herbicide induced adaptive responses. Titers (cfu/mL) were determined at the end of each incubation step on both LB and on Cip at an initially non-permissive concentration (ca. 2×MIC) (*Dan et al., 2016*). The frequency of Cip resistant mutants and the number of generations was calculated.

Experiments described in the section "Herbicide-induced changes in MSC can occur without a change in MIC" were conducted similarly. However, in this case the culture supplemented with only Cip was able to grow because the chosen Cip concentration was below MIC.

### Determination of resistance levels

Individual colonies of strains grown on solid LB medium were used to inoculate 100 µL of liquid LB in a 96 well plate. The plate was incubated with aeration to saturation before ca. 4 µL samples were stenciled onto LB plates containing the appropriate antibiotics. Plates were incubated at 37 °C for 18 h. Strains were scored positively for growth if growth was tangible.

### Mixed culture experiments

Isogenic strains of *E. coli* differing only in the MIC phenotype (i.e., high vs. low) and an additional selection marker encoded on a low copy number plasmid were co-incubated in liquid LB medium containing herbicide, antibiotic, both, or neither and grown to saturation before dilution by a factor of $10^3$ in the same conditions. The antibiotic concentration chosen was below NOEL for both strains. NOEL is defined here as the highest antibiotic concentration for which no effect on survival or selection were observed. The titer of each culture was determined by plating on non-selective medium after each incubation step. After 5 rounds of incubation, the ratio of strains was determined by selecting for the second—competition irrelevant—marker.

Natural selection was defined as the difference in the exponential growth rate of the two strains (*Mallet, 2012*; *Van den Bosch et al., 2014*). Under this interpretation, the change in the proportion of resistant individuals per unit of time is the logistic curve (*Mallet, 2012*) with the explicit solution $p = e^{st}/(c + e^{st})$, where $t$ = time, $p$ = proportion of resistant individuals, $s$ = strength of selection, $c$ = constant describing the initial proportion of resistant individuals, $c = (1/p_0) - 1$. The constant $p_0$ is the proportion of resistant individuals at $t_0$ (the start of the experiment). $t$ is defined as the number of generations, using the generation time for the more resistant strain. By rearranging this formula, the strength of selection is $s = \ln(pc/(1-p))/t$.

### Growth curves

Growth curves were established at 37 °C in liquid LB medium supplemented with herbicide, antibiotic, both, or neither using a FLUOstar Omega microplate reader (BMG LABTECH, Germany). The $OD_{600}$ was determined every 6 min for 16 h and averaged over five biological replicates. Cultures were started at densities of ca. $10^6$ cfu/mL. Antibiotic and herbicide concentrations used are detailed in the legend of Fig. S1.

## Statistical analysis

R was used for all statistical analyses (*R Core Team, 2013*). For experiments measuring changes to MIC, an ANOVA was used to analyze the randomized complete block design, using each independent experiment as a block with presence/absence of antibiotic/herbicide as levels. Residuals were used to test for normality and equality of variances and log transformed data where appropriate. Tukey's HSD test was used to determine which treatments were significantly different from each other.

For mixed culture experiments (changes to MSC), two sets of analyses were performed. First, we determined whether adding antibiotics increases the strength of selection at different herbicide concentrations. Second, we determined whether adding herbicides increases the strength of selection at different antibiotic concentrations. Each set of questions was tested using contrasts performed with the glht function in the multcomp package (*Hothorn, Bretz & Westfall, 2008*) with a two-sided alternative and sequential Bonferroni procedure. These contrasts were fit to an ANOVA model treating each combination of herbicide and antibiotic as a treatment category. Residual plots confirmed that the assumptions of normality and equality of variance were met. Because residuals were approximately normally distributed we report the arithmetic mean (the maximum likelihood estimator of population mean under the assumption of normality).

## Statistical analysis for growth curves

To estimate carrying capacity (k), a logistic growth model on the raw data using non-linear least squares (nls) was fit in R. Differences in population growth between strains in different treatments were estimated by estimating r, the intrinsic growth rate, of each strain. Data sets were log transformed and the slope of the growth curve between $t = 48$ min and $t = 150$ min was measured. A visual inspection of the plots revealed that before $t = 48$ min graphs were not linear. After $t = 150$ min, growth slowed as cells were entering stationary phase.

We tested for differences between the two strains in both r and k by calculating contrast in an ANOVA (aov). Residual plots were used to test for violations of assumptions for ANOVA. Assumptions of normality and equal variances were met in all data sets. In two data sets there were a small number of outliers. These had small influences on parameter estimates and were hence not removed. Contrasts between treatments were calculated using the glht package in R using the sequential Bonferroni correction.

## RESULTS

As reported previously, simultaneous exposure to a variety of antibiotics and either various commercial herbicide formulations (*Kurenbach et al., 2015*) or to either active ingredients and some surfactants (*Kurenbach et al., 2017*), alters the survival of *E. coli* and *S. typhimurium* compared to exposure to only the antibiotics. This was due to adaptive changes in exposed bacteria. From these observations we hypothesized that particular combinations of antibiotics and these formulations alter the MSC or MIC of the antibiotic in a way that favours genetically resistant variants. To test this hypothesis, we conducted

short-term evolution in antibiotics with and without additional exposures to commercial herbicide formulations.

Because the effect of the herbicide exposure was specific to the combination of herbicide and antibiotic, primarily two kinds of experiments were done. Firstly, in the case where the herbicide raised the MIC of the antibiotic, we tested whether antibiotic survivors could give rise to antibiotic resistant variants. In this case, only bacteria simultaneously exposed to both the herbicide and antibiotic would survive when the antibiotic concentration exceeded the MIC of the bacteria measured without the addition of the herbicide. Secondly, in the case where the herbicide decreased the MIC of the antibiotic, we tested whether the herbicide decreased the MSC of the antibiotic, leading to a greater rate of resistance evolution at lower concentrations of antibiotic because of exposure to the herbicide formulation.

## Evolution of acquired antibiotic resistance in cultures with herbicide-induced increases in MIC

Exposing either *E. coli* or *S. typhimurium* to the herbicide formulation Roundup increased the MIC of the fluoroquinolone antibiotic ciprofloxacin, as did exposing *S. typhimurium* to the herbicide formulation Kamba (*Kurenbach et al., 2015*). These three combinations were used to test the hypothesis that the survival of bacteria exposed to normally lethal concentrations of antibiotic provides the opportunity for the population to evolve higher frequencies of resistant genotypes.

The rate of acquired resistance mutations in populations of *E. coli* BW25113 or *S. typhimurium* SL3770 (all strains described in Table 1) was measured over the course of about 25 generations in a standard rich growth medium, liquid LB, or LB+herbicide, LB+Cip, or LB+herbicide+Cip. The chosen herbicide concentration was below NOEL (no observable effect level), so had no impact on survival or selection of the bacteria on its own (Table 2). The ciprofloxacin concentration was the same in all cultures to which it was added and was above MIC in cultures not exposed to herbicide, but below MIC for bacteria in cultures simultaneously exposed to herbicide. At the end of the experiment, bacteria were transferred to solid medium supplemented with high levels of ciprofloxacin (above MIC) and no herbicide, which was permissive only to the growth of variants with acquired resistance.

Expectedly, cultures supplemented only with ciprofloxacin did not grow, and genotypic resistant variants were not detected. Cultures that grew for 25 generations without ciprofloxacin supplementation produced resistant variants at similar low rates regardless of exposure to the herbicide formulations. This indicated that the herbicides were not mutagens at these concentrations. In a separate standard test of mutagenicity (*Funchain et al., 2001*), bacteria were exposed to herbicides and plated on the antibiotic rifampicin. No difference in resistance rates was observed ($p = 0.3873$).

Populations of bacteria with continuous exposure to herbicide and antibiotic had significantly higher numbers of ciprofloxacin resistant variants. The rate ranged from $10^2$ times higher for the combination of Cip+Kamba+*S. typhimurium* to $10^5$ times higher for Cip+Roundup+*E. coli* (Table 2). This is consistent with our prediction that herbicide-induced adaptive resistance that increases MIC allows rare spontaneously arising

**Table 2  Mutation rate (mutant frequency per generation).** Rates of acquired resistance were determined with or without herbicide and Cip exposure. Values are means of at least four independent experiments. To represent sampling uncertainty in the mean value for treatment groups we present standard errors (standard deviation/$\sqrt{n}$) in brackets. Cip concentrations used were 0.07 µg/mL for *S. enterica* and 0.05 µg/mL for *S. enterica* in liquid culture and: 0.07 µg/mL for *S. enterica* and 0.06 µg/mL for *E. coli* for final plating. A total of 1,250 ppm ae Roundup or 1,830 ppm ae Kamba were used.

| | LB | LB+Herbicide | LB+Herbicide+Cip |
|---|---|---|---|
| *S. enterica* | | | |
| Kamba | $3.57 \times 10^{-6} (1.27 \times 10^{-6})$[b] | $2.01 \times 10^{-4} (1.95 \times 10^{-4})$ | $1.30 \times 10^{-2} (1.29 \times 10^{-2})$[c] |
| Roundup | $3.57 \times 10^{-6} (1.27 \times 10^{-6})$[b] | $2.91 \times 10^{-5} (2.47 \times 10^{-5})$ | $2.79 \times 10^{-2} (1.71 \times 10^{-2})$[a,c] |
| *E. coli* | | | |
| Roundup | $1.80 \times 10^{-9} (1.62 \times 10^{-9})$ | $1.97 \times 10^{-10} (5.46 \times 10^{-11})$ | $2.72 \times 10^{-5} (2.67 \times 10^{-5})$[d] |

**Notes.**

[a] Herbicide+antibiotic mutation rates significantly different from LB but not from herbicide treatment, for all other combinations herbicide+antibiotic is significantly different from both LB and herbicide treatments.

[b] *S. enterica* experiments were conducted concurrently, using the same LB controls for both assays. *E. coli* was strain BW25113.

[c] $P < 0.01$.

[d] $P < 0.001$.

ciprofloxacin resistant mutants to increase a culture's MIC. Because ciprofloxacin is mutagenic (*Cirz et al., 2005*), it contributes quantitatively to the genotypic heterogeneity, including acquired antibiotic resistance, among the bacteria that survive because of the effects of the herbicide.

Ciprofloxacin resistant colonies were isolated at the end of each experiment from all treatments. We determined MICs for 56 *S. typhimurium* isolates (isolated on 0.05 µg/mL Cip), 27 isolates from Kamba+Cip and 29 isolates from Roundup+Cip treatments. The parental strain and 2 isolates from each LB, Kamba, and Roundup treatments, also isolated on 0.05 µg/mL Cip, were included as controls.

The parental strain and six evolved isolates did not grow at 0.07 µg/mL, a concentration just above the selection concentration. We observed MICs of 0.1 µg/mL ciprofloxacin for 21 isolates and 0.2 µg/mL ciprofloxacin for 28 isolates. Only seven isolates displayed higher MICs (two from Kamba +Cip, four from Roundup+Cip, and 1 from LB), the highest being 1.25 µg/mL ciprofloxacin reached by one isolate recovered from a Roundup+Cip culture. We observed no correlation between level of resistance and original treatment, which indicates that there are no qualitative differences between the ciprofloxacin resistant variants arising in the different treatments.

## Evolution of acquired antibiotic resistance in cultures with herbicide-induced decreases in MIC

Antibiotic resistant bacteria are becoming a fixed part of many environments despite the concentration of antibiotics often being very low (*Hermsen, Deris & Hwa, 2012*). We hypothesized that exposure to some herbicides can shift the MSC to lower antibiotic concentrations leading to competition between individuals with different physiological responses and thus providing an environment in which genotypically resistant bacteria evolve.

Mixed cultures of *E. coli* were created to represent pre-existing antibiotic resistance heterogeneity within natural environments. The phenotypic differences were created using

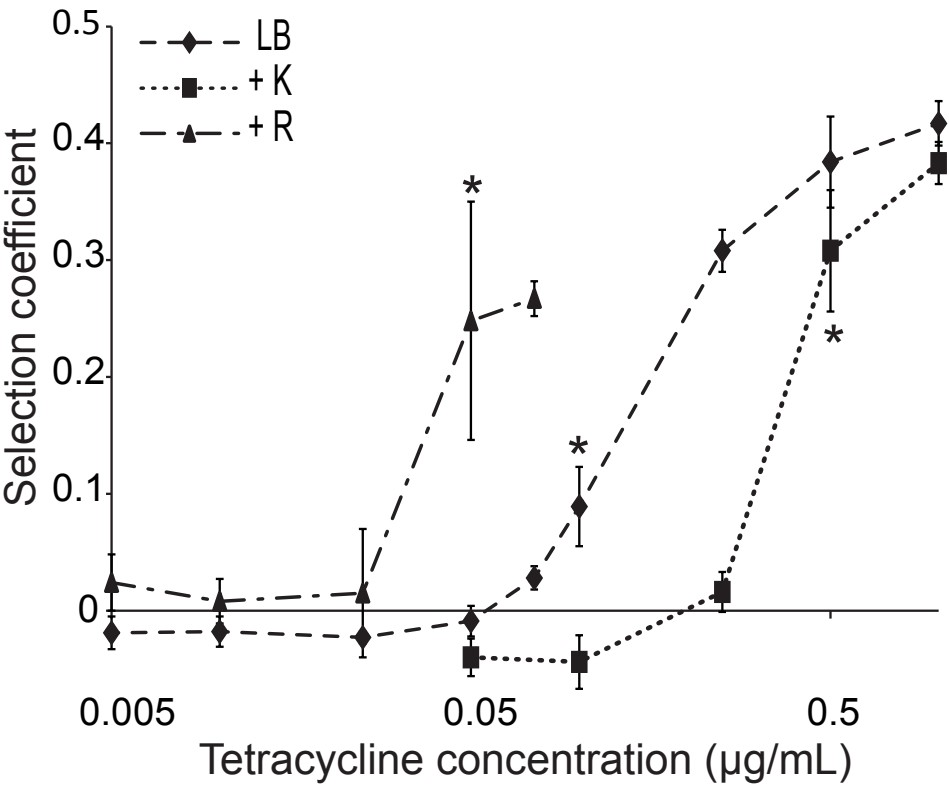

**Figure 2** **Competition between strains of different potential to resist tetracycline.** Competition between tetracycline resistant (AH201, Tet^high) and susceptible (AH214, Tet^low) strains of *E. coli* in different concentrations of tetracycline with or without Kamba (K) or Roundup (R). Kamba was used at 1,830 ppm ae, Roundup at 311 ppm ae. Selection coefficients are presented as a function of tetracycline concentration, with positive values indicating a selection for the more resistant strain. Values are means of three independent experiments. To represent sampling uncertainty in the mean value for treatment groups we present error bars depicting standard errors (standard deviation/$\sqrt{n}$). * indicate the lowest tetracycline concentrations for each treatment where selection coefficients were significantly different from the -Tet/-herbicide treatment, based on a Bonferroni corrected contrast described in the 'Materials and Methods'.

bacteria carrying isogenic plasmids with different alleles of *tetA* (Tet^high, Tet^low, Table 1). The MIC for tetracycline of Tet^low was 1.5 μg/mL and for Tet^high it was 125 μg/mL.

Tet^high and Tet^low were competed in liquid LB, LB+Tet, and LB+Tet+Roundup media for about 42 generations. This herbicide lowers the antibiotic MIC (Fig. 1B). The same concentration of Roundup was used in combination with various concentrations of tetracycline, all of which were below the MIC of Tet^low. Roundup reduced tetracycline MIC for Tet^low to 0.5 μg/mL, but growth of Tet^high was not affected for tetracycline concentrations up to 50 μg/ml, far above levels used in this experiment.

The MSC, i.e., the concentration of tetracycline at which one strain started to dominate the culture, was determined from the calculated selection coefficients. A positive coefficient indicated selection for the more resistant strain. The selection coefficient was significantly different for competitions in environments supplemented with ≥0.1 μg/mL tetracycline compared to competitions in LB medium (Fig. 2). Mixed cultures in LB+Tet+Roundup

**Table 3  Ciprofloxacin resistant variants of *E. coli* BW25113 per generation.** *p*-value for comparison between the treatment condition and LB medium. Kamba was used at 1,830 ppm ae, ciprofloxacin concentration was 0.025 µg/mL in liquid and 0.06 µg/mL in solid media.

| Treatment | Resistant variants/ generation | SEM | *p*-value[a] |
|---|---|---|---|
| LB | $7.5 \times 10^{-9}$ | $3.9 \times 10^{-9}$ | na |
| Kamba | $10^{-7}$ | $6.3 \times 10^{-8}$ | 0.22 |
| Cip | $1.2 \times 10^{-3}$ | $6.5 \times 10^{-4}$ | $2.4 \times 10^{-6}$ |
| Kamba+Cip | $5 \times 10^{-7}$ | $4.5 \times 10^{-7}$ | 0.27 |

had significantly different selection coefficients compared to the LB competitions beginning at a tetracycline concentration of 0.05 µg/mL.

A reverse experiment was performed using Kamba instead of Roundup. This herbicide increases the MIC of Tet$^{low}$ from 1.5 to 3 µg/mL. When Kamba was part of the environment, the selection coefficient was significantly different from the competition in LB medium at a tetracycline concentration $\geq 0.5$ µg/mL, compared to $\geq 0.1$ µg/mL without Kamba (Fig. 2). In both cases, MIC for the Tet$^{low}$ strain and MSC moved in the same direction.

## Herbicide-induced changes in MSC can occur without a change in MIC

In addition to the observed increase or decrease of antibiotic MIC caused by herbicide formulations, we also observed combinations of antibiotic and herbicide that did not change the MIC of the antibiotic but did alter growth rate (Fig. 1C). For example, *E. coli* cultures in medium supplemented with Kamba grew faster at some sub-lethal concentrations of ciprofloxacin, but Kamba did not change the MIC (*Kurenbach et al., 2015*). At the concentrations of the antibiotic at which the herbicide altered growth rate, possibly also the MSC of the antibiotic was changed.

To test this, we measured the frequency at which acquired ciprofloxacin resistance arose during culture in the combination of Cip+Kamba+*E. coli*. The mutation rate of an *E. coli* monoculture was measured over the course of about 25 generations in a standard rich growth medium, liquid LB, and in LB supplemented with Cip, Kamba, or Cip+Kamba. The experiment was similar to those described in the section "Evolution of acquired antibiotic resistance in cultures with herbicide-induced increases in MIC", but here the ciprofloxacin concentration was below MIC and high enough to slow the growth of bacteria in cultures not supplemented with herbicide. At the end of the experiment, bacteria were transferred to solid medium supplemented with ciprofloxacin at >MIC and no herbicide, which was permissive only to the growth of variants with acquired resistance.

Acquired resistance rates were the same for cultures grown in LB, LB+Kamba or LB+Cip+Kamba (for both treatment combinations $p > 0.2$; Table 3). The acquired resistance rate in the LB+Cip culture was about $10^5$ times higher than the other three cultures ($p < 10^{-4}$ for all treatment combinations), indicating that the ciprofloxacin concentration met or exceeded the MSC. The addition of Kamba neutralized the selective effect of ciprofloxacin by shifting the MSC to a higher level without changing the MIC.

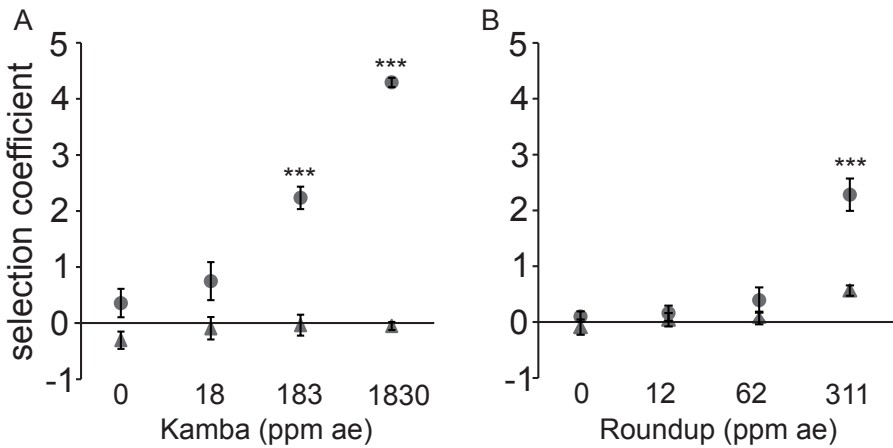

**Figure 3 Dose-response to two herbicides with (●) or without (▲) antibiotic (used at a single concentration).** Dose-response of *E. coli* to two herbicides with (●) or without (▲) antibiotic (used at a single concentration). A positive selection coefficient indicates selection for the strain with a higher MIC. Results are averages of three independent experiments. To represent sampling uncertainty in the mean value for treatment groups we present error bars depicting standard errors (standard deviation/$\sqrt{n}$). (A): Kamba+Str, using strains AH204 (Str$^{high}$, MIC$_{Str}$: 200 µg/mL) and AH211 (Str$^{low}$, MIC$_{Str}$: 1 µg/mL); (B): Roundup+Tet, using strains AH201 (Tet$^{high}$, MIC$_{Tet}$: 125 µg/mL) and AH214 (Tet$^{low}$, MIC$_{Tet}$: 1.5 µg/mL). Concentrations of antibiotics were 0.25 µg/mL for Str and 0.05 µg/mL for Tet. Asterisks indicate contrasts where addition of antibiotic significantly changed the strength of selection, based on a Bonferroni corrected contrast described in the 'Materials and Methods', with *, $p < 0.05$; **$p < 0.01$, ***, $p < 0.001$.

## Herbicide-induced changes to MSC are herbicide concentration-dependent

Pairs of isogenic strains of *E. coli* (Table 1) with either different streptomycin (Str$^{high}$, Str$^{low}$) or tetracycline (Tet$^{high}$, Tet$^{low}$) MICs were competed for about 40 generations. The frequency of each strain after competition was measured. The concentration of the appropriate antibiotic did not vary between cultures, but the concentration of herbicide did. Combinations were chosen that lead to a decrease in MIC with the chosen herbicide, namely Str+Kamba, and Tet+Roundup.

There was no differential in the fitness of paired strains when they were co-cultured in standard rich medium (Fig. 3). In cultures containing antibiotic but not supplemented with herbicide neither strain had a fitness advantage, indicating that the antibiotic concentration was below MSC. Likewise, the herbicide alone did not affect fitness except for a statistically non-significant effect at the highest tested concentration of Roundup.

In each test, the combination of antibiotic and herbicide reduced MSC. The more resistant strain invariably increased relative to the isogenic competitor in a herbicide dose-dependent manner when both antibiotic and herbicide were in the environment. Selection coefficients were statistically significantly different from LB medium at 183 parts per million acid equivalent (ppm ae) Kamba and 311 ppm ae Roundup.

The observed fitness differential was due to the faster reproductive rate of the Str$^{high}$ and Tet$^{high}$ strains in environments with sub-lethal concentrations of antibiotic for both strains in each pair. This was shown by measuring the growth rate of the strains in

monoculture rather than in competition (Fig. S1). No differences were observed for the monoculture growth rate (r) of matched isogenic pairs in LB medium, LB+antibiotic, or LB+herbicide ($p > 0.1$ for all combinations; Figs. S1A and S1C). Only in the combined LB+herbicide+antibiotic treatment did strains with higher MICs have shorter generation times in monoculture than their lower MIC counterparts ($p = 1.6 \times 10^{-5}$ (Kamba+Str) and $p = 4.7 \times 10^{-10}$ (Roundup+Tet)).

Significant differences in the carrying capacity of the environments for all treatment combinations were observed for the matched strains exposed to Kamba+Str measured in monoculture. The high MIC strain achieved higher final optical densities ($p < 2 \times 10^{-16}$; Fig. S1B). In contrast, the Str$^{low}$ strain population grew to higher final optical densities in LB, LB+Str, and LB+Kamba treatments. In cultures with Roundup+Tet, Tet$^{high}$ and Tet$^{low}$ had similar densities ($p = 0.24$, Fig. S1D).

## DISCUSSION

In this study we report that when bacteria are simultaneously exposed to herbicides and antibiotics, mutants with higher levels of resistance can evolve. In some cases, resistance evolved 100,000 times faster.

Herbicides can increase the MIC of some antibiotics. At what otherwise would be a lethal concentration of the antibiotic, the bacteria can continue to reproduce. Each reproductive event has a low but steady potential of producing a variant daughter with a higher MIC. We found that these strains have a fitness advantage and accumulate differentially to their low MIC cousins.

Herbicides can also decrease the MIC of some antibiotics. At what otherwise would be a concentration of antibiotic below the MSC, too low to have an effect on the fitness of two bacteria differing in their MICs, we found that the bacteria with the higher MICs replaced the bacteria with lower MICs. The shift in MSC seemed to be of a similar magnitude as the shift in MIC observed for the lower MIC strain.

Finally, herbicides can also alter survival potential at some antibiotic concentrations but not change the concentration to which the entire population is innately susceptible. In the case presented here, MSC was shifted by the herbicides, but MIC was not. The herbicide mitigated the selective pressure caused by the antibiotic, and no genotypes with higher resistance levels established in the population. Although not tested, it is likely that in the reverse case, when MSC is lowered by a substance, antibiotic resistance may arise at higher frequencies.

Our research shows that manufactured chemical products such as the herbicides can have a complex effect on the evolution of antibiotic resistance. They did not replace antibiotics, but could accelerate resistance evolution. Herbicides that reduce the MSC will be more effective at stimulating resistance evolution at the lower ends of the antibiotic concentration gradient, while herbicides that increase the MIC will be more effective at stimulating resistance evolution at the higher ends.

Antibiotics can influence acquired resistance evolution in a number of ways. First, some antibiotics may also be mutagens (*Cirz et al., 2005*) and at sub-lethal concentrations could

increase the probability of a mutation conferring resistance arising in the population. Second, they discriminate between the reproductive rate of those with higher and lower tolerances to the toxic effects of the antibiotic, leading to different frequencies of genotypes that are less susceptible. At the environmentally relevant concentrations of herbicide formulations that we used, no mutagenic activity was detected. However, the herbicides did contribute to the ability of antibiotics to change the frequency of resistant variants in populations.

Combination treatments using a non-antibiotic to increase an antibiotic's effectiveness have been suggested to help preserve the usefulness of antibiotics (*Allen et al., 2014*; *Schneider et al., 2017*; *Wright, 2016*). Indeed, various combinations are in clinical use (*Worthington & Melander, 2013*). While this might improve or enable treatment of an infection, our results indicate that decreasing bacteria's survival by making an antibiotic more potent may promote resistance evolution by creating more environments in which a resistance genotype is more fit.

Herbicides and different ingredients in product formulations can have antimicrobial activities, with some being more potent antimicrobial agents than are others (*Kurenbach et al., 2017*). For example, Roundup was more toxic than Kamba to the bacteria that we used. The low level toxicity is likely how they induce adaptive resistance, a source of phenotypic heterogeneity (*Kurenbach et al., 2015*). In addition, biocidal agents that are mutagens may introduce genotypic heterogeneity.

Antibiotic and herbicide gradients may be viewed as two environmental dimensions concentrating competition between bacteria of different genotypes and physiotypes. The particular concentrations of antibiotic and herbicide at any point of intersection of the gradients above a threshold level increases the frequency of the most fit strains and species, amplifying any linked or associated traits in the more fit competitor. While antibiotics can do this without herbicides (*Andersson & Hughes, 2014*; *Baquero et al., 1998a*; *Denamur & Matic, 2006*), the herbicides used in this study increased the range of concentrations under which the antibiotic affects the evolution of resistance.

The concentrations of herbicides we used were below recommended application levels and hence within relevant environmental ranges, suggesting that what we observe in the laboratory has the potential to occur in many places where both antibiotics and herbicides are found together. Simultaneous herbicide and antibiotic exposures are common. Herbicides are used in agriculture, where spray drift or walking through treated fields exposes farm livestock and pets, which may be on therapeutic or prophylactic antibiotics. Most ingested antibiotic is not metabolized and thus excreted (*Chee-Sanford et al., 2009*), becoming mixed with soil as crop fertilizer which *in situ* may be subsequently sprayed with herbicide. Microbes from these mixes may be carried by blow- and house-flies (*Zurek & Ghosh, 2014*). Likewise honeybees may be exposed to herbicide spray or residues as they forage and return to an antibiotic-treated hive. Additionally, herbicides are used in urban environments for purposes like gardening and lawn care, including parks and roadsides (*Atwood & Paisley-Jones, 2017*). Worldwide, herbicide use was approximately $1 \times 10^9$ kg in 2012 with up to $2 \times 10^8$ kg of the active herbicidal ingredients glyphosate, 2, 4-D and dicamba used in the US in 2012 (*Atwood & Paisley-Jones, 2017*).

Other chemicals also have been shown to cause adaptive resistance and to increase resistance frequencies (*Egeghy et al., 2012*; *Gustafson et al., 1999*; *Levy, 2001*). Non-antibiotic prescription medicines and food emulsifiers select antibiotic resistant gut bacteria (*Kurenbach et al., 2017*; *Maier et al., 2018*). Approximately 8 million manufactured chemical substances are currently in commerce (*Egeghy et al., 2012*; *Shen, Pu & Zhang, 2011*). According to the US Environmental Protection Agency, annual production of each of the top 3,000 chemicals is greater than $6 \times 10^{11}$ kg/year (*EPA, 2008*). They are not regulated for effects on antibiotic resistance and not tested for such effects.

The susceptibility of bacteria to antibiotics must be seen as a non-renewable resource, one that requires careful stewardship worldwide (*Amabile-Cuevas, 2016*; *Heinemann & Kurenbach, 2017*). Evidence that antibiotic resistance evolution is influenced by exposure of bacteria to a wide range of substances may require us to make changes in how we manage both antibiotics and other manufactured and widely distributed chemical products. This is because many facets of the extrinsic environment induce adaptive changes, a complexity frequently ignored in standard studies of resistance. As our results show, complex effects of exposures to non-therapeutic chemicals may undermine strategies to preserve the effectiveness of antibiotics through altering just their use. To our knowledge, there has been no attempt to systematically test common chemicals to which pathogenic bacteria are chronically exposed for effects on antibiotic resistance.

## CONCLUSIONS

Neither reducing the use of antibiotics nor discovery of new ones may prevent the post-antibiotic era. This is because bacteria may be exposed to other non-antibiotic chemicals that predispose them to evolve resistance to antibiotics more quickly. Herbicides are examples of some of the most common non-antibiotic formulations in frequent global use. More research is necessary to see to what extend other different manufactured chemicals may contribute to this effect. Moreover, depending on how the manufactured chemicals are used, or how they move through the waste stream, there may be combinatorial effects caused by mixtures of different products. Future work should take into account likely combinations as well as different ways that microbes could be exposed to chemical products.

### Abbreviations

| | |
|---|---|
| **MIC** | minimum inhibitory concentration |
| **MSC** | minimum selective concentration |
| **NOEL** | no observable effect level |

## ACKNOWLEDGEMENTS

The authors thank Mark Silby for helpful comments.

### Funding

Amy M. Hill received the New Zealand Federation of Graduate Women and UC for support. This project received funding from the Brian Mason Trust (to Jack A. Heinemann) and donations to the UC Foundation (to Jack A. Heinemann) including from, inter alia, donors Third World Network (Malaysia) and the Sustainable Food Trust (UK). The funders had no role in study design, data collection and analysis, decision to publish, or preparation of the manuscript.

### Grant Disclosures

The following grant information was disclosed by the authors:
New Zealand Federation of Graduate Women and UC.
Brian Mason Trust.
Third World Network (Malaysia).
Sustainable Food Trust (UK).

### Competing Interests

The authors declare there are no competing interests.

### Author Contributions

- Brigitta Kurenbach performed the experiments, analyzed the data, contributed reagents/materials/analysis tools, prepared figures and/or tables, authored or reviewed drafts of the paper, approved the final draft.
- Amy M. Hill performed the experiments, analyzed the data, contributed reagents/-materials/analysis tools, authored or reviewed drafts of the paper, approved the final draft.
- William Godsoe analyzed the data, contributed reagents/materials/analysis tools, authored or reviewed drafts of the paper, approved the final draft.
- Sophie van Hamelsveld performed the experiments, analyzed the data, contributed reagents/materials/analysis tools, approved the final draft.
- Jack A. Heinemann conceived and designed the experiments, analyzed the data, contributed reagents/materials/analysis tools, authored or reviewed drafts of the paper, approved the final draft.

### Data Availability

    The raw data are provided in the Supplemental Files.

### Supplemental Information

Supplemental information for this article can be found online at http://dx.doi.org/10.7717/peerj.5801#supplemental-information.

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
