# Peer review of "Agrichemicals and antibiotics in combination increase antibiotic resistance evolution"

_PeerJ, doi:10.7717/peerj.5801_

## Round 0.1 · original submission · Major Revisions

· Academic Editor

Major Revisions

First, I apologize for the length of time this manuscript was in review. I just could not get the 3rd reviewer to submit their review until now. I do think that the 3 sets of reviewer comments are good, so I would appreciate receiving your response to those comments.
Clarifications on methods are needed as well as some concern about the lack of a control in the experiments.

Reviewer 1 ·

Basic reporting

It is to some extent necessary to read Kurenbacher 2015 and 2017 to get informed on the choice of the study organisms, herbicides and antibiotics and several MIC values. It would ease the reader experience if some more information from these studies were provided to make the rationale for the study and the experimental design more evident. The authors might consider to include MIC values with and without herbicides where relevant in table 1 or in a new table.

Experimental design

With reference to the research question Lines 100-103 “both increases and decreases in MIC caused…”. Why is it not performed experiments on acquired resistance frequencies in drug herbicide combinations that reduced the MIC. This is only explored exclusively in MSC assays with defined pre-existing genetic variation. From figure 1 in Kurenbach et al 2015 it looks like there is several candidate combinations to explore this. Can we assume that this will cause the reverse result of the experiment performed with antibiotic/herbicide combinations resulting in an increased MIC? Also see point 1 in “Validity of results”

Validity of the findings

The experiments can broadly be divided into two different types of assays that is used to test various hypotheses, short term evolution in the form of fitness assays and mutation frequency assays. I have two larger concerns regarding the mutation frequency assays.

1. Line 193 – 229 and Table 2: It is not clear to me how the authors can claim that acquired resistance increase due to herbicide-induced increases in MIC without designing this assay in a manner allowing for frequency data on LB + CIP (below MIC) as a control in this assay. Was the MSC for LB+CIP+Roundup/Kamba higher than the MIC of the parental strain? In its current form the results obtained from this assay looks somewhat forced since selective pressure on resistance development in essence only was allowed in the CIP+Herb combinations.

2. The innate mutagenic effect of ciprofloxacin is not tested nor discussed even though that can be considered an alternative explanation for -or bias the results seen in the mutation frequency assays. The higher mutation frequencies reported on LB+CIP in Table 3 compared to LB+CIP+Herbicide in Table 2 might indicate this. See for instance Cirz et al 2005 plos biol, https://doi.org/10.1371/journal.pbio.0030176 were this issue is addressed.

Additional comments

I struggle to keep the flow through the results section. For instance, Fig 2 and Fig 3B is clearly linked sharing a significant data point applying the exact same environment. However, there is a whole chapter with different methods between the two assays. It would be nice if the authors could try and bridge the different sections better so it is easier to follow.

Annotated reviews are not available for download in order to protect the identity of reviewers who chose to remain anonymous.

Reviewer 2 ·

Basic reporting

The manuscript needs additional information to better relay the experimental design and results to the public. There are questions regarding methods that need to be addressed. See comments to authors section.

Experimental design

The experimental design was not clearly articulated. More detail is in the comments to authors section.

Validity of the findings

The reviewer isn't convinced that comparison of bacterial growth curves after exposure to a mixed chemical formulation warrants statements that attribute endpoints to the active ingredient in those formulations.

Additional comments

Review of ms # 27947 – Agrichemicals boost the effects of antibiotics on antibiotic resistant evolution

First thoughts… This is a very broad title that seems to encompass myriad chemical formulations used by the agricultural complex and then goes further to promise how this knowledge applies to (I’m assuming at this point) bacterial evolution. So, big promises.

-on to the abstract-

First notes:

“Antibiotic resistance is medicine’e climate change…”. Hmmm…

Why use single quotes around ‘adaptive’? Sounds like there is a caveat coming.

Next sentence starts with “Adaptive” yet no explaination of ‘adaptive’.

Line 26 – “…of the antibiotic”. What antibiotic? If you haven’t introduced “the antibiotic” then maybe say, “Adaptive changes can increase fitness under selective pressures such as those presented by antibiotics”.

Line 36 – This reviewer prefers to stay away from idioms such as “damned if you do/damned if you don’t”, plus I’m not sure it really applies here?

Line 49/50 – “…accelerate genotypic resistance…” This manuscript does not provide any information regarding genetic underpinnings of antibiotic resistance.

Line 67 – Is there a better way to say “invent us away from”?

Line 68 – replace “using them less” with “reduction of use” and the rest of the sentence with “which will help increase longevity”.

Lines 86-88 – “…resistance through changes in gene expression is also known as an adaptive (change in phenotype) response.” The reviewer disagrees with this interpretation of adaptive, or at least believes non-heritable acclimation by changes in gene expression are just that, acclimation. While heritable changes in genetic composition that allow for the persistence of a phenotype in a population are adaptive.

Materials and Methods
Media
The authors have used commercial herbicide formulations containing either a chlorinated benzoic acid, dicamba, or the organophosphate, glyphosate as Kamba and Roundup, respectively. Throughout the manuscript they attribute their observations to the active ingredient, however, the emulsifiers and detergents in these formulations may have profound effects on bacteria.

Plasmid constructs
The authors should use this section to better explain how the plasmids were utilized to reach their end goals. This would benefit from the addition of the plasmid maps as a supplemental figure. What did each trait allow? For example, simple addition of explanatory text such as, “which encoded for resistance to the antibiotics chloramphenicol, ampicillin, and tetracycline, respectively.” to the current sentence, “The resulting traits for this plasmid were Cam, Amp, and Tet.”, yields the sentence, “The resulting traits for this plasmid were Cam, Amp, and Tet, which encoded for resistance to the antibiotics chloramphenicol, ampicillin, and tetracycline, respectively.” This help the reader that may not be intimately familiar with the subject material to better understand the context.

Culturing conditions
Why differing seeding densities? Most likely because this allowed the best resolution of growth curves, but does seeding density have an effect on growth?

Determination of resistance levels
More information is needed here. Media concentrations are not listed.

What effects were measured to contrive the NOEL used here. Why NOEL vs. MIC as the rest of the paper?

Line 149 – Natural selection was defined as… then the author cited an article by van den Bosch et al. 2014. When the reviewer browsed the article, one of the first paragraphs that grabbed attention was, “Our findings are generalizations for fungal plant pathogens and fungicides only. They do not necessarily apply to resistance against antibiotics, insecticides, or herbicides because of the differences in genetic systems and the reproduction biologies of the species groups involved…” The authors should at least explain why they think this is a good approach in spite of the warning in their citation.

There are many other instances where this manuscript should be re-worked.

·

Basic reporting

I found this quite a difficult paper to review as it took quite a bit of rereading of sections to get their meaning. I think the introduction needs a little more background as to the findings of their previous papers on the subject. You really have to have read those for this to make any sense. At times in the results section, it was also unclear whether the authors were referring to their previous work or to the data in the current paper.

I also have specific comments on the presentation of the data - means vs medians, SEMs vs SD, but include these under the 'General comments to the author' section of the review. For those not familiar with the way the data is presented, I think the figure and table legends would benefit from a short explanation of why the data is presented that way.

Experimental design

The research fits within the scope of PeerJ, and the research question is well defined, relevant and meaningful. However, I felt that the methods were not described in enough detail and that there were important controls not included in the experimental design. For many of the experiments, there was no control of the effect of the antibiotics alone on mutation frequencies. There was also no control for whether it was the active ingredient of the herbicides that have the effect, or whether it was other components like surfactants and other ingredients. While I understand that the authors may not think those controls relevant seeing as the herbicides are used as is, as a reader I wanted to know more about what might be responsible.

Validity of the findings

I found it difficult to assess validity without the controls I described previously.

Additional comments

Line 21: Abstract – “Antibiotic resistance is medicine’s climate change: caused by human activity” – This is a really strong and powerful statement and I like the sentiment, but it is a little misleading as currently written as antibiotic resistance itself is a natural phenomenon. Suggest the authors change to something that reflects the growing crisis caused by resistance organisms rather than using the term ‘antibiotic resistance’.
Lines 66-67: I’m not sure I agree with the sentiment that “Despite over half of century of warning, neither science nor innovation has managed to invent us away from the threat of a post-antibiotics era”. This suggests that we’ve had half a century of science/innovation actively working to invest us away from the threat, but that is clearly untrue. We’ve had half a century of inaction, which is not the same thing at all.
Line 82: missing word – ‘can also dependent’
Line 146: Please define NOEL
Line 160: biological or technical replicates?
Lines 208-209. The authors state that “Genetic variants able to grow on high concentrations of ciprofloxacin after 25 generations in LB medium, with or without herbicide supplementation, arose at the same frequency.” I’m a bit confused about this statement. Does this relate to the data presented in Table 2, because that doesn’t seem to be what the data shows, at least for Kamba and S. enterica. Our have I misunderstood that table?
Line 219: should : be ,?
Table 2: Please justify use of mean over median and of SEM over standard deviation (SD), and indicate what statistical tests performed. My understanding is that the SD indicates the dispersion of individual data values around their mean, and is what is required here for the reader to fully understand the variation in the data presented. In contrast, the SEM is a measure of the variability of the means that would be expected if the study were exactly replicated many more times and is used to calculate confidence intervals so isn’t appropriate here (see: http://ww1.cpa-apc.org/Publications/Archives/PDF/1996/Oct/strein2.pdf). What is the frequency of acquired resistance to Cip alone for comparison? This table isn’t very clear to me. I’m confused by which are significantly different from each other. I wonder if this data would be better presented as dot plots of the individual data points for each independent experiment. Is the a that relates to statistically testing in the right place in the table?
Figure 2: As before, please justify use of SEM over SD and indicate what statistical tests performed. Please also include name of bacteria being tested here.
Figure 3: Please indicate if mean or median values are given. As before, please justify use of mean over median and SEM over SD, and indicate what statistical tests performed. Please also include name of bacteria being tested here.

---

## Round 0.2 · Minor Revisions

· Academic Editor

Minor Revisions

Reviewer 1 has some additional points that I'd like to ask you to address. I hope you are willing to do so.

Reviewer 1 ·

Basic reporting

I'm satisfied with the authors corrections.

Experimental design

Satisfied with the author response

Validity of the findings

Regarding my concerns about the lack of mutation rate data on CIP only treatment (point 5 -7):

In the previous version the heading was, in my opinion, formulated in a manner that would have required this assay to have been designed in a way that allowed for mutation rates on CIP-only treatment. With the new heading this is no longer a necessity. I fully understand that the LB-CIP functioned as a control in your experiments when designed like this.
Yes, the experiments answer the hypothesis that herbicide induced increase in MIC increase the likelihood of resistance development of cip resistant mutants in an above wildtype MIC cip containing environment.
It is nice that you now mention mutagenic effects of antibiotics in the discussion but I still believe should be clearer that the several-fold increase in mutation rates might be due to the mutagenic effect of cip in a cip containing environment. This is because this assay cannot distinguish between selective effects and inherent mutagenic effects. My point is that if you used a non-mutagenic antibiotic substance you might not have seen a similarly elevated mutation rate when compared to the LB or LB-Herb treatment.

Additional comments

The authors make a big point in that they are measuring mutation rates and not frequencies. However, these are still also in the new version used interchangeably in the manuscript. See line
180 - For mutant frequency experiments
255 - The frequency ranged from

legend text table 2 (Frequencies of acquired resistance…; Herbicide+antibiotic mutant frequency…

Satisfied with all other responses and corrections made by the authors.

Reviewer 2 ·

Basic reporting

The manuscript is well-written, and clear. The authors have improved the flow of the manuscript from the first version so that it is now even more available to scientists that may not be experts in this field.

Experimental design

Data acquisition and analyses are sound.

Validity of the findings

The conclusions of this manuscript are valid and give credence to the concern that combinations of chemicals can affect the acquisition of, at least acclimative, resistance in bacteria.

Additional comments

Thank you to the authors for taking the time to revise the manuscript. This reviewer thinks that, overall, the manuscript has been greatly improved during the process.

·

Basic reporting

The authors have addressed all of my queries/comments to my satisfaction.

Experimental design

The authors have addressed all of my queries/comments to my satisfaction.

Validity of the findings

The authors have addressed all of my queries/comments to my satisfaction.

Additional comments

Thanks for addressing all of my comments/queries. I'm still not sure I agree with your reworded assertion re having not innovated ourselves out of the antimicrobial resistance crisis, I'm happy to agree to disagree on this point.

---

## Round 0.3 · accepted · Accept

· Academic Editor

Accept

Thanks for your patience and efforts in revising your manuscript.

#